# Improvement of Non-Hydrostatic Hydrodynamic Solution Using a Novel Free-Surface Boundary Condition

**Augusto Hugo Farias Cunha** [1,*] **, Carlos Ruberto Fragoso, Jr.** [2] **, Cayo Lopes Bezerra Chalegre** [1] **and David Motta-Marques** [1]

1   Instituto de Pesquisas Hidráulicas, Universidade Federal do Rio Grande do Sul, Porto Alegre, RS 91501-970, Brazil; cayo.chalegre@ufrgs.br (C.L.B.C.); dmm@iph.ufrgs.br (D.M.-M.)
2   Centro de Tecnologia, Universidade Federal de Alagoas, Maceió, AL 57072-970, Brazil; ruberto@ctec.ufal.com
*   Correspondence: hugo.cunha@ufrgs.br; Tel.:+55-51-3308-6670

**Abstract:** Hydrodynamic models based on the RANS equation are well-established tools to simulate three-dimensional free surface flows in large aquatic ecosystems. However, when the ratio of vertical to horizontal motion scales is not small, a non-hydrostatic approximation is needed to represent these processes accurately. Increasing efforts have been made to improve the efficiency of non-hydrostatic hydrodynamic models, but these improvements require higher implementation and computational costs. In this paper, we proposed a novel free-surface boundary condition based on a fictional sublayer at the free-surface (FSFS). We applied the FSFS approach at a finite difference numerical discretization with a fractional step framework, which uses a Neumann type of boundary condition to apply a hydrostatic relation in the top layer. To evaluate the model performance, we compared the Classic Boundary Condition Approach (CBA) and the FSFS approach using two numerical experiments. The experiments tested the model's phase error, capability in solving wave celerity and simulate non-linear wave propagation under different vertical resolution scenarios. Our results showed that the FSFS approach had a lower phase error (2 to 5 times smaller) than CBA with a little additional computational cost (ca. 7% higher). Moreover, it can better represent wave celerity and frequency dispersion with 2 times fewer layers and low mean computational cost (CBA $\delta$t = 2.62 s and FSFS $\delta$t = 1.22 s).

**Keywords:** non-hydrostatic pressure; implementations cost; computational cost

## 1. Introduction

Hydrodynamic models based on the Reynolds Averaged Navier-Stokes (RANS) equation are well-established tools to simulate three-dimensional free surface flows in large aquatic ecosystems, such as lakes, estuaries, reservoirs, and coastal zones [1–5]. These models usually are based on the hydrostatic assumption of the pressure distribution, which is applied satisfactorily to large shallow water ecosystems with relatively low computational cost algorithms [6]. However, when the ratio of vertical to horizontal motion scales is not small, a non-hydrostatic approximation is needed to model accurately three-dimensional free surface flows.

Few alternatives were explored to improve efficiency and reduce the computational cost of non-hydrostatic hydrodynamic models. Most of them are dedicated to improving the model's ability to solve the elliptic equation to non-hydrostatic pressure through more suitable boundary conditions and use a vertical momentum discretization less dependent on the velocity vertical profile [7–12].

Some approaches use a hybrid model (hydrostatic and non-hydrostatic), minimizing the computational cost by identifying regions where the hydrostatic assumption may be applied [6,13,14]. Optimization in source code processing is also a feasible solution, like a parallel computational algorithm, which allocated the loop calculations on the distributed threads [6,15]. The implementation costs are associated with these issues and aspects related to assumptions adopted in models.

As for the assumptions, two different types of boundary conditions may be applied for non-hydrostatic pressure at the free-surface: (a) a homogenous Dirichlet condition, where pressure is set equal zero at the free-surface (e.g., [6,7,16,17]); and (b) a Neumann condition, usually a hydrostatic relation to approximate pressure's value equal to zero at the free-surface (e.g., [14,18–20]). Both types of free-surface boundary conditions were well discussed in Bergh and Berntsen [21]. However, many numerical models solve Poisson's equation for pressure at the center of the computational cell (e.g., [8,11,14,22,23]). This assumption, without a suitable treatment, may incorrectly estimate the non-hydrostatic boundary condition to the center of the top layer instead of at the free-surface. This assumption becomes more inaccurate as the distance between the center of the top layer and the free surface increases. Hence, a high vertical resolution (1–20 layers) is needed to achieve accurate results for different studies case [7–11,24].

In addition to using a suitable vertical discretization, [8,9] showed that cell-center non-hydrostatic models might have a substantial accumulated phase error, due to the wrong computations in wave celerity. Many efforts were carried out in order to properly overcome this issue, of which we can highlight: (a) an implementation of edge-based non-hydrostatic pressure (e.g., [7,25]), (b) the use of an integration method to estimate the pressure at the free-surface based on pressure in the center of the top layer (e.g., [8,10,11]), and (c) the use of a piecewise linear profile of the non-hydrostatic pressure to estimate the pressure at the free-surface (e.g., [12]). For the existing cell-centered models, the previous algorithms may demand a substantial implementation cost due to changes in the vertical momentum discretization and in the treatment needed to address the non-hydrostatic pressure at the free-surface boundary condition adequately. Thus, in this paper, we proposed a low implementation cost method to approach the free-surface non-hydrostatic pressure in hydrodynamics models properly. The technique is a novel free-surface boundary condition based on a fictional sublayer at the free-surface (*FSFS*), to reach satisfactory results without changing the momentum equations.

We applied the FSFS, and the CBA approaches at a finite difference discretization with a fractional step framework based on [23], which uses a Neumann type of boundary condition that applies a hydrostatic relation in the top layer. To evaluate the model performance, we used two widely applied numerical models benchmarks [8,11,22,23,26–28], selected to test our algorithm in two different purposes: (a) a standing wave in a three-dimensional closed basin to test the model's capability in solving wave celerity under different vertical resolutions; and (b) a wave propagation over a submerged bar to validate the proposed boundary condition at the free-surface and evaluated the effect of vertical resolution under a non-linear wave propagation.

## 2. Methods

### 2.1. Governant Equations

The RANS equations are used to describe three-dimensional free-surface flows. These equations express the physical principle of volume, mass, and momentum conservation. The momentum equations for an incompressible fluid have the following form:

$$
\begin{aligned}
\frac{\partial u}{\partial t} &+ u\frac{\partial u}{\partial x} + v\frac{\partial u}{\partial y} + w\frac{\partial u}{\partial z} - fv = -\frac{\partial p_a}{\partial x} - g\frac{\partial \eta}{\partial x} \\
&- g\frac{\partial}{\partial x}\left[\int_z^\eta \frac{\rho - \rho_0}{\rho_0}d\zeta\right] - \frac{\partial q}{\partial x} + v^h\left(\frac{\partial^2 u}{\partial x^2} + \frac{\partial^2 u}{\partial y^2}\right) + \frac{\partial}{\partial z}\left(v^v\frac{\partial u}{\partial z}\right),
\end{aligned}
\tag{1}
$$

$$\frac{\partial v}{\partial t} + u\frac{\partial v}{\partial x} + v\frac{\partial v}{\partial y} + w\frac{\partial v}{\partial z} - fu = -\frac{\partial p_a}{\partial y} - g\frac{\partial \eta}{\partial y}$$
$$- g\frac{\partial}{\partial y}\left[\int_z^\eta \frac{\rho - \rho_0}{\rho_0}d\zeta\right] - \frac{\partial q}{\partial y} + v^h\left(\frac{\partial^2 v}{\partial x^2} + \frac{\partial^2 v}{\partial y^2}\right) + \frac{\partial}{\partial z}\left(v^v\frac{\partial v}{\partial z}\right),$$

(2)

$$\frac{\partial w}{\partial t} + u\frac{\partial w}{\partial x} + v\frac{\partial w}{\partial y} + w\frac{\partial w}{\partial z} = -\frac{\partial q}{\partial z} + v^h\left(\frac{\partial^2 w}{\partial x^2} + \frac{\partial^2 w}{\partial y^2}\right) + \frac{\partial}{\partial z}\left(v^v\frac{\partial w}{\partial z}\right),$$

(3)

where $u(x,y,z,t)$, $v(x,y,z,t)$, and $w(x,y,z,t)$ are the velocity components in the horizontal ($x$ and $y$) and vertical ($z$) directions ($m/s$), respectively; $v^h$ e $v^v$ are the horizontal and vertical turbulent eddy viscosity coefficients ($m/s$), respectively; $t$ is the time ($s$); $p_a(x,y,z,t)$ is the atmospheric pressure normalized by fluid density ($Pa.m^3/kg$); $\eta$ is the free surface elevation ($m$) from a water level References; $q(x,y,z,t)$ denotes the non-hydrostatic pressure component normalized by fluid density ($Pa.m^3/kg$); $f$ is the Coriolis parameter ($s^{-1}$); and $g$ is the gravitational acceleration ($m/s^2$). The second and third terms on the right-hand side of Equations (1) and (2) represent the barotropic and the baroclinic contributions to the hydrostatic pressure.

When a simple hydrostatic approach is considered, Equation (3) is neglected and q is assumed to be equal to zero in Equations (1) and (2). In this case, it is assumed that vertical acceleration do not have a significant effect in the velocity field in comparison with horizontal acceleration.

The volume conservation is expressed by the incompressibility condition and the continuity equation, given by:

$$\frac{\partial u}{\partial x} + \frac{\partial v}{\partial y} + \frac{\partial w}{\partial z} = 0.$$

(4)

Integrating Equation (4) over depth yields to the following equation:

$$\int_{-h}^\eta \left[\frac{\partial u}{\partial x} + \frac{\partial v}{\partial y} + \frac{\partial w}{\partial z}\right]dz = \int_{-h}^\eta \frac{\partial u}{\partial x}dz + \int_{-h}^\eta \frac{\partial v}{\partial y}dz + \int_{-h}^\eta \frac{\partial w}{\partial z}dz = 0,$$

(5)

where "$h$" is the bathymetry measured from the theoretical undisturbed water surface (zero referential). Using the Leibniz integration rule, in each direction, in Equation (5) and using a kinematic condition at the free-surface leads to the following free-surface equation:

$$\frac{\partial \eta}{\partial t} + \frac{\partial}{\partial x}\int_{-h}^\eta udz + \frac{\partial}{\partial y}\int_{-h}^\eta vdz = 0,$$

(6)

The boundary conditions were implemented under the assumption of "free-slip" boundaries. The Dirichlet and Neumann conditions were assigned to represent the normal and tangential velocities in the solid boundaries, respectively.

The tangential stress boundary conditions for the momentum equations (Equations (1) and (2)) are specified at the free-surface by the prescribed wind stresses, which can be approximated as:

$$v^v\frac{\partial u}{\partial z} = \gamma_T(u_a - u)\,,\ v^v\frac{\partial v}{\partial z} = \gamma_T(v_a - v),\quad at\quad z = \eta,$$

(7)

where $u_a$ and $v_a$ are the horizontal wind velocity components, and $\gamma_T$ is a non-negative wind stress coefficient. The bottom friction is specified by:

$$v^v\frac{\partial u}{\partial z} = \gamma_B u\,,\ v^v\frac{\partial v}{\partial z} = \gamma_B v,\quad at\quad z = -h,$$

(8)

where $\gamma_B$ is a non-negative bottom friction coefficient.

*2.2. Grid and Variables Locations*

The computational grid can be described as a generic unstructured orthogonal grid, having a set of non-overlapping convex $N_p$ elements, each having an arbitrary number of sides $S_i \geq 3, i = 1, 2, \ldots, N_p$

(Figure 1). Let $N_s$ be the total number of sides in the grid. The length of each side is $\lambda_j$, $j = 1, 2, \ldots, N_p$. The vertical faces of the $i$-th element are identified by an index $j_{(i,l)}$, $l = 1, 2, \ldots, S_i$, so that $1 \leq j_{(i,l)} \leq N_s$. Similarly, the two polygons which share the $j$-th vertical face of the grid are identified by the indices $i_{(j,1)}$ and $i_{(j,2)}$, so that $1 \leq i_{(j,1)} \leq N_p$ and $1 \leq i_{(j,2)} \leq N_p$. The nonzero distance between centers of two adjacent polygons which share the $j$-th side is denoted with $\delta_j$.

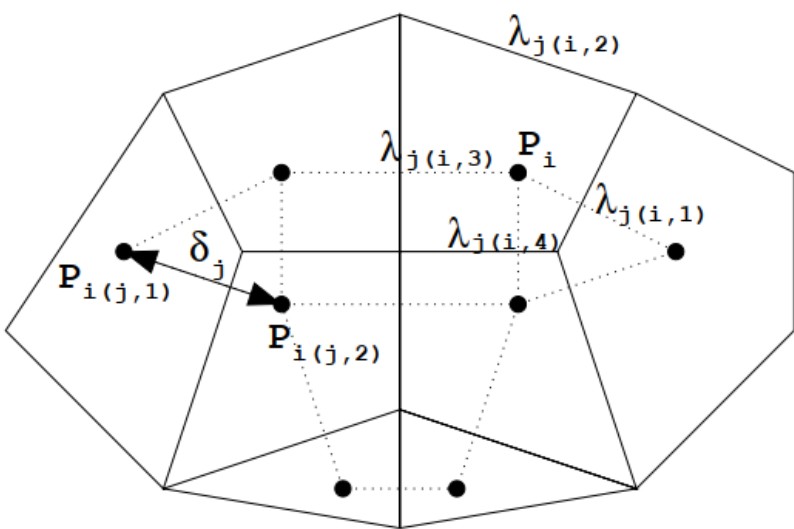

**Figure 1.** Model representation of the grid. (Source: Casulli and Lang [23]).

Along the vertical direction, a simple finite difference discretization, not necessarily uniform, is adopted. By denoting with $\Delta z_{k+\frac{1}{2}}$ a given top computational cell level surface, the vertical discretization step is defined by:

$$\Delta z_k = \Delta z_{k+\frac{1}{2}} - \Delta z_{k-\frac{1}{2}} \quad k = 1, 2, \ldots, N_s. \tag{9}$$

The three-dimensional spatial discretization consists of elements whose horizontal faces are the polygons of a given orthogonal grid, represented by the layers at $k + \frac{1}{2}$ (upper face) or $k - \frac{1}{2}$ (bottom face), whose height, for each layer, is $\Delta z_k$. The water surface elevation, $(\eta)$, is located at the barycenter of the upper horizontal face for each $i$-th element. The velocity component normal to each horizontal face is assumed to be constant over the face of each computational cell and is defined at the point of intersection between the face and the segment joining the centers of the two prisms which share the face. The non-hydrostatic pressure component $q_{i,k}^n$ is located at the center of the $i$-th computational cell, halfway between $\Delta z_{k+\frac{1}{2}}$ and $\Delta z_{k-\frac{1}{2}}$. Finally, the water depth $h_j$ is specified and assumed constant on each vertical face of an element.

### 2.3. Numerical Approximation

We used a semi-implicit method of finite volume ($\theta$-Method [29]) with an Eulerian–Lagrangian Method [30] to solve the convective and viscous terms of the RANS equations, which applies a quadratic interpolation [31] to estimate the velocity field at the end of backtracking process (multi-step backward Euler with ten sub-time steps). A fractional-step framework [23] is used to solve the pressure component by splitting into hydrostatic and non-hydrostatic parts. The algorithm procedures take the following steps:

1. Definition of initial parameters, initial conditions, and boundary conditions.
2. Solution of convective terms using the Eulerian–Lagrangian Method.
3. Determination of the provisional free-surface elevation ($\tilde{\eta}$) through the preconditioned conjugate gradient iterations until the residual norm is smaller than a given tolerance $\epsilon_q$.

4. Numeric solution of provisional velocity field ($\tilde{u}$ and $\tilde{w}$).
5. Solution of non-hydrostatic pressure ($q$) through the preconditioned conjugate gradient iterations until the residual norm is smaller than a given tolerance $\epsilon_q$.
6. Numeric correction of velocity field and free surface elevation.

A complete description of the numerical solution may be found in [23]. Here we focus only in described the conventional non-hydrostatic pressure discretization separately, the given boundary condition, and how applied the *FSFS* boundary condition in the conventional solution to improve the non-hydrostatic hydrodynamic solution.

2.3.1. Non-Hydrostatic Pressure Discretization

A correction of provisional velocity field ($\tilde{u}_{j,k}^{n+1}, \tilde{w}_{j,k}^{n+1}$) are computed after including the non-hydrostatic pressure terms, specifically:

$$u_{j,k}^{n+1} = \tilde{u}_{j,k}^{n+1} - \theta \frac{\Delta t}{\delta_j} \left( \tilde{q}_{i(j,r),k}^{n+1} - \tilde{q}_{i(j,l),k}^{n+1} \right), \tag{10}$$

$$w_{j,k}^{n+1} = \tilde{w}_{j,k}^{n+1} - \theta \frac{\Delta t}{\Delta z_{i,k+\frac{1}{2}}} (\tilde{q}_{i,k+1}^{n+1} - \tilde{q}_{i,k+1}^{n+1}), \tag{11}$$

where the vertical space increment $\Delta z$ is defined as the distance between two consecutive level surfaces, except near the bottom and near the free surface where $\Delta z$ is the distance between a level surface and the bottom, or free-surface, respectively. The $\tilde{q}$ term denotes the non-hydrostatic pressure term, which in combination with the provisional free-surface elevation ($\tilde{\eta}$), gives the pressure:

$$p_{j,k}^{n+1} = g \left( \tilde{\eta}_i^{n+1} - z_k \right) + \tilde{q}_{i,k}^{n+1}, \tag{12}$$

where $z_k$ is the z-coordinate of the $k$-th horizontal level surface, and $g$ is the gravity acceleration. In each computational cell bellow the free-surface, the finite volume discretized incompressibility condition is taken to be:

$$\sum_{L=1}^{S_i} s_{i,L} \lambda_{j(i,L)} \Delta z_{j(i,L),k}^n u_{j(i,L),k}^{n+1} + V_i \left( w_{i,k+\frac{1}{2}}^{n+1} - w_{i,k-\frac{1}{2}}^{n+1} \right) = 0 \quad k = m, m+1, \ldots, M-1, \tag{13}$$

where $V_i$ is the area of the $i$-th polygon and $S_i$ is the number of sides for the $i$-th element. At the free-surface, the finite difference approximation of Equation (6) considering $w_{i,m-\frac{1}{2}}^{n+1} = 0$ and using the incompressibility condition (13) is:

$$V_i \eta_i^{n+1} = V_i \eta_i^n - \theta \Delta t \sum_{l=1}^{S_i} \left[ s_{i,l} \lambda_{j(i,L)} \Delta z_{j(i,l),M}^n u_{j(i,l),M}^{n+1} \right] + \theta \Delta t V_i w_{i,M-\frac{1}{2}}^{n+1}$$
$$- (1 - \theta) \Delta t \sum_{l=1}^{S_i} \left[ s_{i,l} \lambda_{j(i,l)} \sum_{k=m}^{M} \Delta z_{j(i,l),k}^n u_{j(i,l),k}^n \right], \tag{14}$$

where $\theta$ is the implicitness factor, $s_{i,l}$ is sign function associated with the orientation of the normal velocity defined on the $l$ side of an element $i$. Assuming that the pressure at the FSFS is hydrostatic, the pressure correction term ($\tilde{q}_{i,M}^{n+1}$) is obtained by the following hydrostatic relation:

$$p_{j,M}^{n+1} = g \left( \eta_i^{n+1} - z_M \right) = g \left( \tilde{\eta}_i^{n+1} - z_M \right) + \tilde{q}_{i,k}^{n+1}, \tag{15}$$

hence, a substitution the $p_{j,M}^{n+1}$ in the Equation (12) yields:

$$V_i\widetilde{\eta}_i^{n+1} = gV_i\left(\eta_i^n - \widetilde{\eta}_i^{n+1}\right) - g\theta\Delta t \sum_{L=1}^{S_i}\left[s_{i,L}\lambda_{j(i,L)}\Delta z_{j(i,L),M}^n u_{j(i,L),M}^{n+1}\right] + g\theta\Delta t V_i w_{i,M-\frac{1}{2}}^{n+1}$$

$$- g\left(1-\theta\right)\Delta t \sum_{L=1}^{S_i}\left[s_{i,L}\lambda_{j(i,L)}\sum_{k=m}^{M}\Delta z_{j(i,L),k}^n u_{j(i,L),k}^n\right]. \tag{16}$$

A system of equations to solve non-hydrostatic pressure is now derived by substituting the expressions for the new velocities from (10) and (11) into (13) and (16), respectively. The following finite difference equations are obtained:

$$g\theta^2\Delta t^2 \left[\sum_{l=1}^{S_i}s_{i,L}\lambda_{j(i,l)}\Delta z_{j(i,l),k}^n \frac{\widetilde{q}_{i[j(i,l),1],k}^{n+1} - \widetilde{q}_{i[j(i,l),2],k}^{n+1}}{\delta_{j(i,l)},k} + V_i\left(\frac{\widetilde{q}_{i,k}^{n+1} - \widetilde{q}_{i,k+1}^{n+1}}{\Delta z_{i,k+\frac{1}{2}}^n} - \frac{\widetilde{q}_{i,k-1}^{n+1} - \widetilde{q}_{i,k}^{n+1}}{\Delta z_{i,k-\frac{1}{2}}^n}\right)\right]$$

$$= g\theta\Delta t V_i\left(\widetilde{w}_{i,k-\frac{1}{2}}^{n+1} - \widetilde{w}_{i,k+\frac{1}{2}}^{n+1}\right) - g\theta\Delta t \sum_{l=1}^{S_i}s_{i,L}\lambda_{j(i,l)}\Delta z_{j(i,l),k}^n \widetilde{u}_{j(i,l),k}^{n+1}, \quad k = m, m+1, \ldots, M-1, \tag{17}$$

$$g\theta^2\Delta t^2 \left[\sum_{l=1}^{S_i}s_{i,L}\lambda_{j(i,l)}\Delta z_{j(i,l),M}^n \frac{\widetilde{q}_{i[j(i,l),1],M}^{n+1} - \widetilde{q}_{i[j(i,l),2],M}^{n+1}}{\delta_{j(i,l),M}} - V_i\frac{\widetilde{q}_{i,M-1}^{n+1} - \widetilde{q}_{i,M}^{n+1}}{\Delta z_{i,M-\frac{1}{2}}^n}\right] + V_i\widetilde{q}_{i,M}^{n+1} =$$

$$g\theta\Delta t V_i\widetilde{w}_{i,M-\frac{1}{2}}^{n+1} - g\theta\Delta t \sum_{l=1}^{S_i}s_{i,L}\lambda_{j(i,l)}(\Delta z_{FSFS})\widetilde{u}_{j(i,l),M}^{n+1} + gV_i(\eta_i^n - \widetilde{\eta}_i^{n+1}) -$$

$$g(1-\theta)\Delta t \sum_{l=1}^{S_i}[s_{i,L}\lambda_{j(i,l)}\sum_{k=m}^{M}\Delta z_{j(i,l),k}^n u_{j(i,l),k}^n, \quad k = M. \tag{18}$$

Once the non-hydrostatic pressure terms are computed, the velocities field are corrected by Equation (10), while vertical velocity can be estimated, equivalently, by Equation (11) or by the incompressibility condition (13) by setting $w_{i,m+\frac{1}{2}}^{n+1} = 0$:

$$w_{i,k+\frac{1}{2}}^{n+1} = w_{i,k-\frac{1}{2}}^{n+1} - \frac{1}{V_i}\sum_{L=1}^{S_i}s_{i,L}\lambda_{j(i,L)}\Delta z_{j(i,L),k}^n u_{j(i,L),k}^{n+1}; \quad k = m, m+1, \ldots, M-1. \tag{19}$$

This equation guarantees that the resulting velocity field is exactly discrete divergence-free [23], so we used it to compute the vertical velocity components.

The final free surface elevation is obtained by the hydrostatic relation (15) as follows:

$$\eta_i^{n+1} = \widetilde{\eta}_i^{n+1} + \frac{\widetilde{q}_{i,M+1}^{n+1}}{g}. \tag{20}$$

Finally, the non-hydrostatic pressure component can be obtained by:

$$q_{i,k}^{n+1} = \widetilde{q}_{i,k}^{n+1} - \widetilde{q}_{i,M+1}^{n+1}; \quad k = m, m+1, \ldots, M, M+1, \tag{21}$$

making the non-hydrostatic pressure at free-surface equal to zero (i.e., for $k = M+1$).

### 2.3.2. Free-Surface Boundary Condition Treatment

To apply the Fictional Sublayer at the Free-Surface (*FSFS*) into the existing computational domain is only required one additional numerical vertical layer at the top of the computational domain, which does not account to the computational domain. As the height of the *FSFS* is assumed to be equal to zero, the model can always be guaranteed the hydrostatic relation at the free-surface, independent of the number of vertical layers. As the pressure is estimated at the center of the layer,

which can be further away from the surface the smaller the number of layers in CBA, the proposed FSFS condition solves these problems and provides a more physically consistent numerical solution, since the non-hydrostatic pressure condition is set in the fictional sublayer at the free-surface. To use this method only is required a simple adaption in the numerical Equations (17) and (18), considering the position of the layer:

(i) For bottom and middle layers (i.e., $k = m$ to $M - 1$), the Equation (17) is applied using its original form;

(ii) For the top layer ($k = M$), Equation (17) is adapted to take into account the influence of FSFS height ($\Delta z_{FSFS}$) in $\Delta z^n_{i,M+\frac{1}{2}}$ and $\Delta z^n_{j(i,l),k}$:

$$
g\theta^2 \Delta t^2 \left[ \sum_{l=1}^{S_i} s_{i,L}\lambda_{j(i,l)} \left( \Delta z^n_{j(i,l),M} - \Delta z_{FSFS} \right) \frac{\widetilde{q}^{n+1}_{i[j(i,l),1],M} - \widetilde{q}^{n+1}_{i[j(i,l),2],M}}{\delta_{j(i,l)}, M} + \right.
$$
$$
\left. \left[ V_i \left( \frac{\widetilde{q}^{n+1}_{i,M} - \widetilde{q}^{n+1}_{i,M+1}}{\Delta z^n_{i,M+\frac{1}{2}}} - \frac{\widetilde{q}^{n+1}_{i,M-1} - \widetilde{q}^{n+1}_{i,M}}{\Delta z^n_{i,M-\frac{1}{2}}} \right) \right] \right] = g\theta\Delta t V_i \left( \widetilde{w}^{n+1}_{i,M-\frac{1}{2}} - \widetilde{w}^{n+1}_{i,M+\frac{1}{2}} \right) -
$$
$$
g\theta\Delta t \sum_{l=1}^{S_i} s_{i,L}\lambda_{j(i,l)} \left( \Delta z^n_{j(i,l),M} - \Delta z_{FSFS} \right) \widetilde{u}^{n+1}_{j(i,l),M}, \quad k = M,
\tag{22}
$$

$$
z^n_{i,M+\frac{1}{2}} = \frac{1}{2} \left[ \left( \Delta z^n_{i,M} - \Delta z_{FSFS} \right) + \Delta z_{FSFS} \right] = \frac{1}{2} \left( \Delta z^n_{i,M} \right)
\tag{23}
$$

$$
\Delta z^n_{i,M-\frac{1}{2}} = \frac{1}{2} \left[ \left( \Delta z^n_{i,M} - \Delta z_{FSFS} \right) + \Delta z^n_{i,M-1} \right];
\tag{24}
$$

(iii) For the FSFS layer ($k = M + 1$), Equation (18) is adapted to take into account the *FSFS* height and the velocity field in the layer $M$. Preliminary simulations showed that making $\Delta z_{FSFS} = 0$ a stable solution is achieved for any vertical discretization:

$$
g\theta^2 \Delta t^2 \left[ \sum_{l=1}^{S_i} s_{i,L}\lambda_{j(i,l)} \left( \Delta z_{FSFS} \right) \frac{\widetilde{q}^{n+1}_{i[j(i,l),1],M+1} - \widetilde{q}^{n+1}_{i[j(i,l),2],M+1}}{\delta_{j(i,l)}, M} - V_i \frac{\widetilde{q}^{n+1}_{i,M} - \widetilde{q}^{n+1}_{i,M+1}}{\Delta z^n_{i,(M+1)-\frac{1}{2}}} \right] + V_i \widetilde{q}^{n+1}_{i,M+1} =
$$
$$
g\theta\Delta t V_i \widetilde{w}^{n+1}_{i,M-\frac{1}{2}} - g\theta\Delta t \sum_{l=1}^{S_i} s_{i,L}\lambda_{j(i,l)} \left( \Delta z_{FSFS} \right) \widetilde{u}^{n+1}_{j(i,l),M} + gV_i(\eta^n_i - \widetilde{\eta}^{n+1}_i)
$$
$$
- g(1-\theta)\Delta t \sum_{l=1}^{S_i} [s_{i,L}\lambda_{j(i,l)}] \sum_{k=m}^{M} \Delta z^n_{j(i,l),k} u^n_{j(i,l),k}, \quad k = M + 1,
\tag{25}
$$

where

$$
\Delta z^n_{i,(M+1)-\frac{1}{2}} = \frac{1}{2} \left[ \left( \Delta z^n_{i,M} - \Delta z_{FSFS} \right) + \Delta z_{FSFS} \right] = \frac{1}{2} \left( \Delta z^n_{i,M} \right).
\tag{26}
$$

## 2.4. Numerical Experiments

The proposed numerical approach was applied in two consolidated benchmarks, usually used to verification and validation of numerical models (e.g., [8,11,22,23,26,28]). Each numerical experiment has a different purpose, as follows:

a Standing waves in a three-dimensional closed basin: This test was widely applied in the literature to verify the model's ability to simulate 3D linear waves comparing the analytic solution with the numerical solution in regard to phase and amplitude representation [8,9,11,20]. We evaluated the model capability in calculating the wave celerity and frequency wave dispersion with the Classic Boundary Approach (hereafter named CBA) and with the proposed FSFS boundary condition. We used six different vertical resolutions (20~5 layers), as most of the previous studies do, and since previous analyses showed that more than 20 layers do not have substantial improvement over

20 layers non-hydrostatic solution. We compare the free surface elevation cumulative phase error, the mean one time-step computational cost, the number of wave periods, and the relation with the free surface vertical velocity after 30 s of simulation in comparison with the analytical solution. We also compared the free surface elevation results with some metrics (RMSE, BIAS, Volume Error, KGE, NSE). At last, we statistically tested the residual series (the difference between analytic and simulated results) with the non-parametric Kruskal–Wallis test followed by a post-hoc Nemenyi to identify significant differences concerning the analytic results. The mean time of one time-step simulation was computed using an Intel® Xenon® CPU-E5-1620 3.7 GHz computer with 32 GB of RAM in a Fortran-based numerical model.

b The wave propagation over a submerged bar: This test case was an experimental model idealized by [32], and was frequently used to validate numerical models (e.g., [7,8,24,26,28,33]). This experiment was used to evaluate accuracy to represent a non-linear wave pattern due to physical changes at the bottom, by comparing free surface elevation between the FSFS approach with a different vertical resolution between simulated and experimental results. To evaluate the model's performance was used a few metrics (RMSE, BIAS, Volume Error, KGE, NSE) and the statistical test non-parametric Kruskal–Wallis test followed by a post-hoc Nemenyi test applied at residuals series.

## 3. Results

### 3.1. 3D Standing Waves in a Closed Basin

In this test, was analyzed a flow induced by an initial wave amplitude set to 0.1 m in a closed cubic basin with 10 m of edge (Figure 2). To discretize the spatial domain, was adopt a regular with 0.5 m resolution, resulting in 8.000 computational cells. The simulation is carried along 30 s, with a time-step size $\Delta t = 0.01$ s. The analytic solution of free-surface water elevation is given by:

$$\eta = A \cos (k_x x) \cos (k_y y) \cos \left( 2\pi \frac{t}{T} \right), \tag{27}$$

where t is the time (the initial condition in the free surface may be obtained by doing $t = 0$); T is the wave period equal to 3.1 s, with the wave number $kx = ky = n/L$ with the total wave number $k = \sqrt{k_x^2 + k_y^2} = 0.44 \frac{rad}{m}$.

The analytic solution for each velocity component is described as follows:

$$u = \frac{Agk_x}{\omega} \frac{\cosh [k_x (h + z)]}{\cosh (k_x h)} \sin (k_x x) \cos (k_y y) \sin (\omega t), \tag{28}$$

$$u = \frac{Agk_y}{\omega} \frac{\cosh [k_y (h + z)]}{\cosh (k_y h)} \cos (k_x x) \sin (k_y y) \sin (\omega t), \tag{29}$$

$$w = \frac{Agk_x}{\omega} \frac{\sinh [k_x (h + z)]}{\cosh (k_x h)} \cos (k_x x) \cos (k_y y) \sin (\omega t), \tag{30}$$

where $\omega$ represents the wave dispersion relation, given by:

$$\omega = \sqrt{gK \tanh (Kh)}. \tag{31}$$

The study case applied different vertical resolution scenarios (i.e., 20, 16, 13, 10, 8, and 5 vertical layers), with both *CBA* and the proposed *FSFS* condition (called methods). We evaluated the cumulative phase error of a 3D standing wave by comparing the model outcomes (i.e., free-surface elevation and free-surface vertical velocity component) at $x = y = 0.25$ m with the analytical solution. The performance between boundary condition approaches was evaluated by comparing the free-surface elevation residuals. The performance between scenarios (considering a different

number of vertical layers) was also evaluated by statistically tested the residues with a non-parametric Kruskal–Wallis test followed by a Nemenyi test since the residues series do not follow a normal distribution (Shapiro–Wilk test). The numeric experiment aims to identify a critical vertical resolution for the FSFS method, where the results are significantly different from the used best scenario for this benchmark (20 vertical layers).

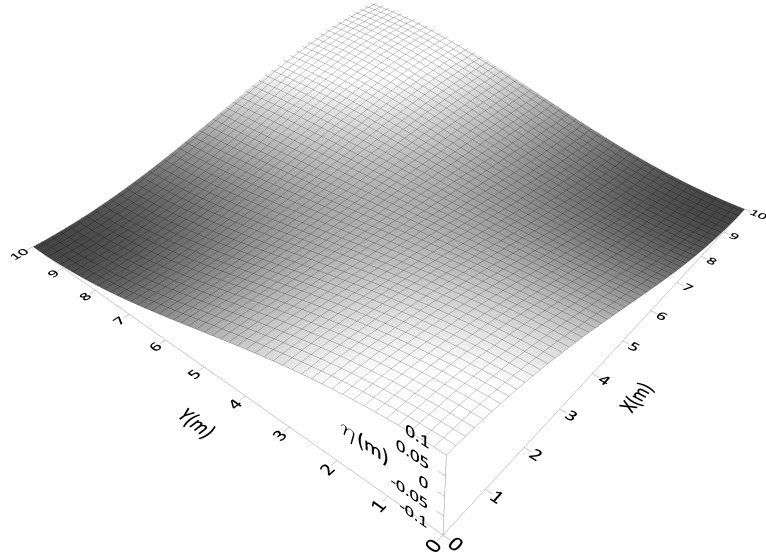

**Figure 2.** The initial free-surface profile for a linear 3D standing wave oscillation in a closed basin. (Source: Yuan and Wu [8]).

In general, for both methods, the phase error ($\Phi_\varepsilon$) increases over time step, and it becomes bigger with the reduction of the number of layers (Table 1). For simulations CBA approach, these errors are higher; for instance, the 20-layer CBA simulation had comparable results with the 8-layers FSFS simulation (Figure 3). Furthermore, when phase error is critical, a decrease in the cumulative free surface elevation error in the low vertical resolution results occurs (less than ten) due to an increase in wave periods. Due to this, results with fewer layers appeared better than those with more layers, as can be seen by comparing FSFS-5L with FSFS-10L results (Figure 3). These effects complicate the direct comparison between methods; however, the cumulative residual free surface elevation error (Figure 4) can clarify the matter. Due to this, the calculated metrics (Table 2 and 3) was only comparable between 0 and 10 s of simulations (Figure 4). This effect can be better verified in CBA accumulated residual series when the graphic changes the slope, which occurs sooner as the number of layers reduced (Figure 4).

**Table 1.** Computational cost, Phases Error, and the number of wave periods between different methods and scenarios. The model was implemented with Fortran and simulated in a machine using an Intel R Xenon R CPU-E5-1620 3.7 GHz computer with 32 GB of RAM.

| NºL | FSFS | | | CBA | | |
|---|---|---|---|---|---|---|
| | $\Delta t(s)$ | Nº T | $\Phi_\varepsilon$ | $\Delta t(s)$ | Nº T | $\Phi_\varepsilon$ |
| 20-Layers | 2.8 | 10 | 0.3 | 2.62 | 10 | 1.6 |
| 16-Layers | 2.21 | 10 | 0.4 | 2.14 | 10 | 2 |
| 13-Layers | 1.68 | 10 | 0.6 | 1.67 | 10 | 2.4 |
| 10-Layers | 1.54 | 10 | 1 | 1.45 | 11 | 3.1 |
| 8-Layers | 1.51 | 10 | 1.5 | 1.34 | 11 | 3.8 |
| 5-Layers | 1.22 | 11 | 2.1 | 1.08 | 12 | 4.6 |

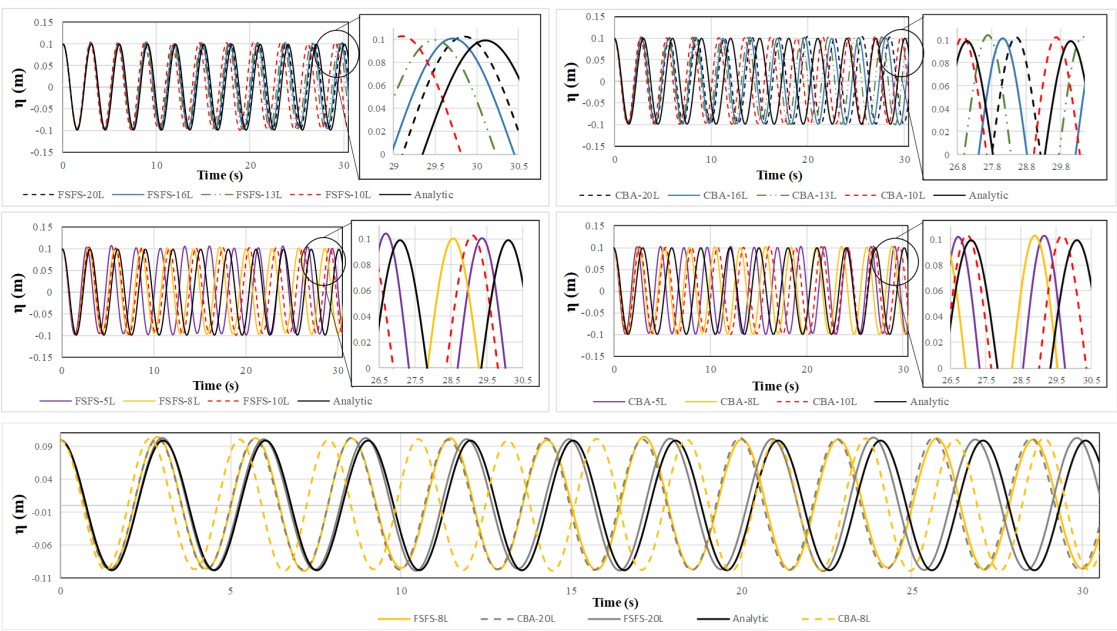

**Figure 3.** Free surface elevation at $x = y = 0.25$ m for 30 s of simulation: (**top**) Comparing analytic solution with simulated solution for 20 to 5 layers scenario with fictional sublayer at the free-surface (*FSFS*) condition (left side); (**middle**) Comparing analytic solution with simulated solution for 20 to 5 layers scenario with Classic Boundary Condition Approach (*CBA*) condition (right side); and (**bottom**) Comparing both methods thought the 20 layer scenario and 8 layers scenario (at bottom).

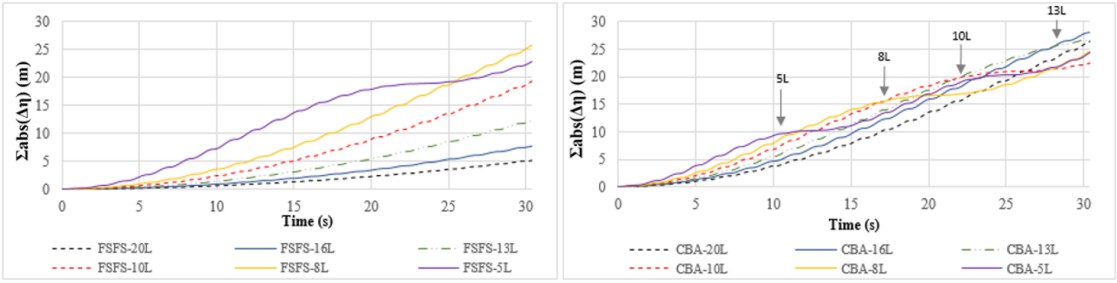

**Figure 4.** Free surface elevation accumulated residuals series for the FSFS approach (**left**) and CBA approach (**right**), at x = y = 0.25 m for 30 s of simulation, comparing different layers scenarios.

**Table 2.** Metrics between the analytic and simulated results from the FSFS method for each scenario for the first 10 s of simulation.

| Metrics | FSFS-20L | FSFS-16L | FSFS-13L | FSFS-10L | FSFS-8L | FSFS-5L |
|---|---|---|---|---|---|---|
| RMSE (mm) | 7.40 | 11.52 | 18.29 | 30.53 | 45.30 | 89.20 |
| BIAS (mm) | 0.32 | 0.32 | −0.77 | −2.11 | −3.54 | −7.61 |
| Error (%) | 9.25 | 14.28 | 22.55 | 37.79 | 56.60 | 114.13 |
| KGE | 0.93 | 0.93 | 0.82 | 0.52 | 0.18 | −0.88 |
| NSE | 0.99 | 0.97 | 0.93 | 0.81 | 0.57 | −0.66 |

All statics metrics indicated a decrease of performance with a reduction of the number of layers. The metrics show that CBA-20L simulation had similar results as the FSFS-8L, as well as CBA-10L to FSFS-5L. The CBA method also had a decrease of performance with a reduction of vertical resolution (Tables 3 and 4). When CBA method was used, the phase error was 2 to 5 times higher, and with little difference in computational cost (ca. 7% smaller) (Table 1) in comparison with the *FSFS* method. We found a phase error similar to previous works using 20-layers with *CBA* method [8,9]. The statistical

test for the FSFS approach showed significant differences between low (below 10 layers) and high (above 13 layers) vertical resolution, considering the free surface elevation residuals. Thus, simulation with 20 to 10 layers has similar (see Table 4) and acceptable results (Tables 1 and 2).

**Table 3.** Metrics between the analytic and simulated results from the CBA method for each scenario for the first 10 s of simulation.

| Metrics | CBA-20L | CBA-16L | CBA-13L | CBA-10L | CBA-8L | CBA-5L |
|---------|---------|---------|---------|---------|--------|--------|
| RMSE (mm) | 47.06 | 57.68 | 68.75 | 83.48 | 95.63 | 107.80 |
| BIAS (mm) | −4.35 | −5.65 | −6.88 | −7.96 | −7.57 | −1.06 |
| Error (%) | 58.94 | 72.70 | 87.10 | 106.42 | 123.39 | 145.02 |
| KGE | 0.004 | −0.30 | −0.61 | −0.91 | −0.93 | −0.20 |
| NSE | 0.54 | 0.31 | 0.017 | −0.45 | −0.90 | −1.42 |

**Table 4.** Nemenyi posthoc test comparing the FSFS residue series of the simulation with 20 to 5 vertical layers to identify the significative statistical difference between results.

| Nº-L | FSFS-20L | FSFS-16L | FSFS-13L | FSFS-10L | FSFS-8L |
|------|----------|----------|----------|----------|---------|
| FSFS-16L | 0.97833 | - | - | - | - |
| FSFS-13L | 0.24554 | 0.69401 | - | - | - |
| FSFS-10L | < 0.05 | < 0.05 | 0.05269 | - | - |
| FSFS-8L | < 0.05 | < 0.05 | < 0.05 | < 0.05 - | |
| FSFS-5L | < 0.05 | < 0.05 | < 0.05 | 0.70396 | 0.5515 |

Our findings indicated that the accuracy of the vertical velocity component is affected by the vertical resolution applied, influencing the representation of the wave phase directly (Figure 3). For the FSFS method, we have noted that the magnitude of vertical velocity components increases or decreases depending on the vertical discretization used, since the 10 layers setup a threshold value (Figure 3). For the CBA approach, the magnitude of vertical velocity components decreases above 16, leading to a less flexible critical vertical resolution. Moreover, we also observed that the variation in horizontal resolution did not affect wave phase representation, which seems to be more directly related to the wave damping (not analyzed in this paper).

### 3.2. Wave Propagation over a Submerged Bar

A scheme of the experiment of the wave propagation over a submerged bar with an uneven bottom may be seen in (Figure 5) [32]. At the upward slope of the bar, the shoaling wave becomes non-linear due to the generation of bound higher harmonic. At the downward slope, the depth increases rather fast, and these harmonics become free, resulting in an irregular pattern of waves [26]. The numerical reproduction of this pattern has been shown to be very demanding in terms of the accuracy of the computed dispersion frequency [7].

The computational domain has a total length of 30 m, with an initial undisturbed water level of 0.4 m, which was discretized using a regular grid of 0.025 m resolution. The simulation is carried along 39 s, with a time-step size $\Delta t = 0.005$ s. At the left boundary, a sinusoidal wave condition, with period $T = 2$ s and amplitude $A = 0.01$ m, was imposed to represent the wave generator of the original experiment. At the right outflow boundary, the experimental absorbed beach was computationally represented by a $5m - sponge$ layer with a combination of a sponge layer technique [34] and a Sommerfeld-type radiation boundary condition, applied to minimize wave reflection, given by:

$$\epsilon_i = \begin{cases} \beta \left( \frac{x_i - x_{io}}{l_i} \right)^2 \left( \frac{z_m - z}{z_m - z_M} \right) u_i & if \quad x_i \geq x_{io} \\ 0 & if \quad x_i < 0 \end{cases} , \tag{32}$$

where $\epsilon_i$ is the sponge layer coefficient; $x_{i0}$ is the initial point; $l_i$ the total length. This term must be added in the right side of Equations (1) and (2).

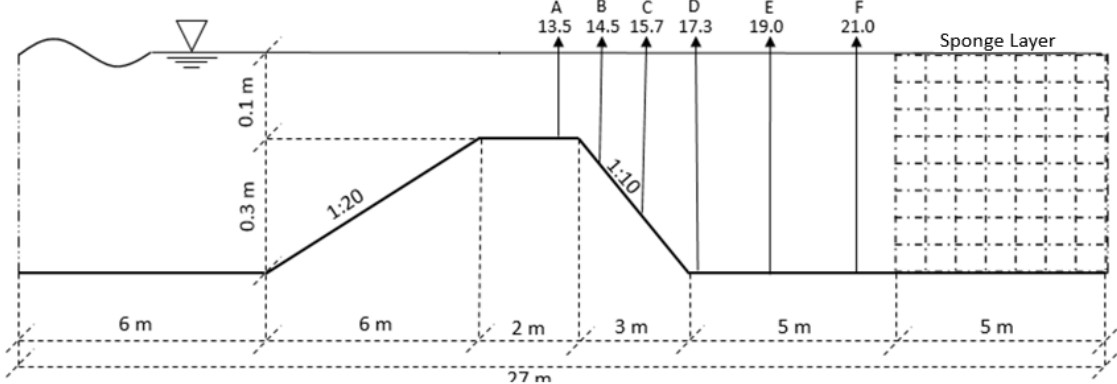

**Figure 5.** Scheme of experimental bottom geometry and location of wave level gauges. (Source: Beji and Battjes [32]).

We evaluated the capability of the model in simulating the dispersion properties of the flow, comparing simulated and measured free-surface elevation between 33 and 39 seconds of simulation under different vertical discretization scenarios. We used the proposed *FSFS* approach since the first numerical experiment showed that the *CBA* method only might reach similar results to the *FSFS* approach using a higher number of layers (Figure 3). The *FSFS* approach was tested for 20, 16, 10, and 5 vertical layers comparing the free surface elevation at the six stations with the measurements of free-surface elevation obtained experimentally by Beji and Battjes [32].

The results (Figure 6) indicated that the non-linearity after the upward slope (a), at the beginning and the middle of downward slope (b–c) were well represented by vertical resolutions from 20 to 10 layers, by comparing the patterns of the experimental results of Higher Amplitude Waves (HW) (e.g., 36.5 s station b) and Low Amplitude Waves (LW) (e.g., 36 s station a). For the stations (d), (e), and (f), after the deshoaling process, the simulations using 10, 16, and 20 layers were less accurate in representing the non-linearity pattern. Specifically, the LW in the station (d) (e.g., 35 to 36 s) and the HW in stations (e) and (f) (36 s at station f). The 5-layers scenario showed lower performance in comparison with the other scenarios, with an oscillatory behavior for the HW in stations (a) and (b), substantial phase errors (station c), and low capacity to represent LW and HW amplitudes (stations e and f).

The FSFS results (Table 5) were capable of satisfactorily representing the phase, amplitude, and wave pattern for all stations, using a higher vertical resolution (20 to 10 layers with FSFS method) (NSE from 0.94 to 0.5). Although, the results had higher RMSE and Bias (between 13.7 and −9.9) relative to the mean maximum and minimum amplitudes for the six stations (4.1 and −6.3 mm) and had high Volume Error (between 26% and 116%), always underestimating the free-surface level, and with low performance in KGE parameter due to the increased Volume Error. As we can see in Figure 6, the high Volume Error is expected. Moreover, the Kruskal–Wallis and Nemenyi statistical tests did not indicate a significant difference between the residues series, except in station "A" for 5 layers scenario.

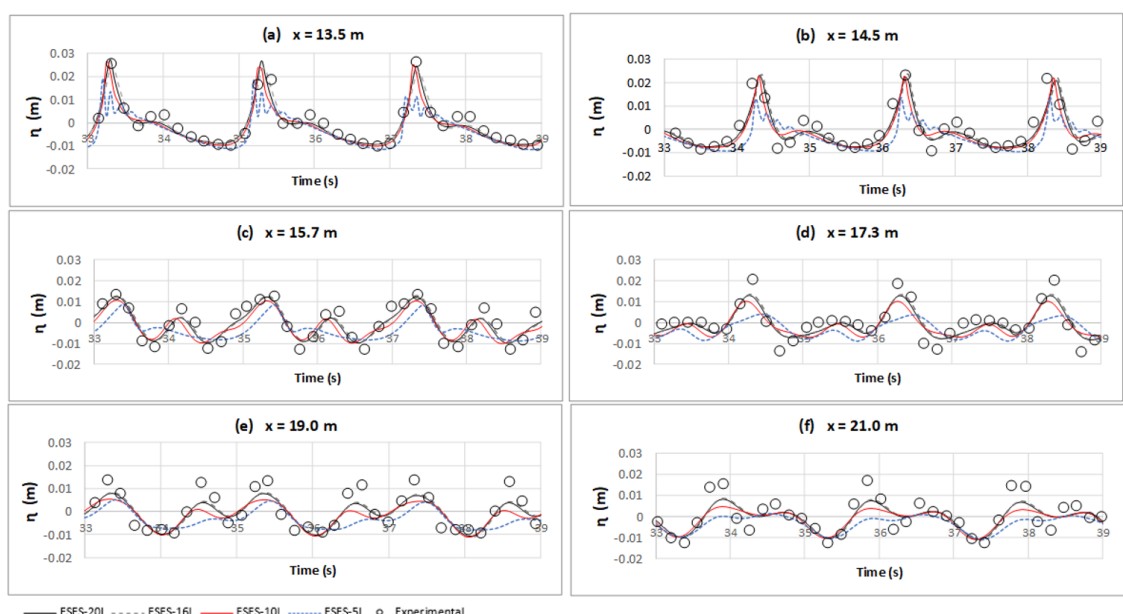

**Figure 6.** Comparisons between experimental (circles) and computed data with 20-layers (solid black), 16-layers (dashed gray), 10-layers (solid red), and 5-layers (dashed blue), at 6 different level gauges.

**Table 5.** Statistics metrics between simulated and experimental results for the six stations for each used layer scenario with the FSFS method.

| Metrics | Station a: x = 13.5 m | | | | Station d: x = 17.3 m | | | |
|---|---|---|---|---|---|---|---|---|
| | **20L** | **16L** | **10L** | **5L** | **20L** | **16L** | **10L** | **5L** |
| RMSE (mm) | 2.67 | 2.31 | 3.40 | 5.23 | 4.12 | 4.08 | 5.57 | 8.05 |
| BIAS (mm) | −0.67 | −0.70 | −1.26 | −2.70 | −0.60 | −0.44 | −1.34 | −3.14 |
| Error (%) | 26.47 | 25.37 | 34.07 | 53.94 | 59.29 | 57.97 | 79.03 | 116.84 |
| KGE | −0.95 | −1.06 | −2.69 | −6.92 | −4.57 | −3.06 | −11.30 | −27.85 |
| NSE | 0.92 | 0.94 | 0.87 | 0.69 | 0.76 | 0.76 | 0.56 | 0.08 |
| | Station b: x = 14.5 m | | | | Station e: x = 19.0 m | | | |
| RMSE (mm) | 3.68 | 4.21 | 4.18 | 6.74 | 4.30 | 4.23 | 5.69 | 7.22 |
| BIAS (mm) | −0.80 | −0.73 | −1.23 | −2.58 | −0.65 | −0.50 | −1.39 | −3.28 |
| Error (%) | 37.73 | 43.83 | 44.30 | 76.27 | 48.15 | 47.35 | 62.91 | 75.29 |
| KGE | −0.83 | −0.68 | −1.80 | −4.88 | −54.45 | −41.24 | −117.00 | −277.78 |
| NSE | 0.82 | 0.77 | 0.77 | 0.41 | 0.72 | 0.72 | 0.50 | 0.20 |
| | Station c: x = 15.7 m | | | | Station f: x = 21.0 m | | | |
| RMSE (mm) | 3.91 | 3.64 | 5.45 | 7.64 | 4.57 | 4.59 | 5.61 | 7.09 |
| BIAS (mm) | −0.63 | −0.48 | −1.37 | −3.32 | −0.65 | −0.48 | −1.38 | −3.25 |
| Error (%) | 44.46 | 40.98 | 62.63 | 88.06 | 56.63 | 56.12 | 67.62 | 76.43 |
| KGE | −1.13 | −0.63 | −3.58 | −10.09 | −9.80 | −7.09 | −22.02 | −53.11 |
| NSE | 0.79 | 0.82 | 0.60 | 0.21 | 0.69 | 0.68 | 0.53 | 0.24 |

## 4. Discussion

The statement that "algorithms which define non-hydrostatic pressure at the center of free-surface computational cell need 10 to 20 vertical layers to resolve wave frequency dispersion to an acceptable accuracy" was wildly reported to justify the use or the proposal of new approaches to the free-surface boundary conditions for non-hydrostatic pressure and momentum equation discretization [7–11,24]. In general, this vertical resolution limitation is addressed to a classic finite difference discretization with

cell center non-hydrostatic pressure without boundary treatment at the free-surface computational cell (e.g., [23]).

The improvement in non-hydrostatic hydrodynamic solutions is due to the capability to set the value of dynamic pressure close to zero at the free-surface instead of the top layer, for any vertical resolution using a FSFS condition. In a classic boundary approach, as expected, the calculated non-hydrostatic pressure in the top layer increased with the reduction of the number of layers. This occurs because, in this approach, the non-hydrostatic pressure is estimated at the center of the layer, which can be further away from the surface as the number of layers becomes smaller. When using the proposed FSFS condition, the special treatment that occurs in the upper layer minimizes these effects and provides a more physically consistent numerical solution, since this component disappears into real flows.

The results showed that the wave phase representation of short waves in deep water conditions ($\frac{H}{\lambda} > 0.5$ m) is related to the contour condition for the non-hydrostatic pressure at the free-surface, but also is influenced by vertical momentum discretization applied. Our findings showed that the proposed boundary condition was capable of improving the model capability in solving wave celerity and wave frequency dispersion, in relation to the classic boundary condition approach. Also, the proposed boundary conditions have a substantial reduction in the computation effort to reach similar results (ca. 146% *FSFS* 8 layers and *CBA* 20 layers), and also a little additional computational effort (ca. 7%) when compared to the same benchmark setup, though with a much higher performance (comparing Tables 2 and 3).

Besides the boundary condition treatment, the phase representation is also related to the used vertical momentum discretization less dependent on the velocities vertical profile. The number of layers may affect the vertical velocity estimative, thus affecting the non-hydrostatic pressure estimation and, hence, the free-surface elevation and velocity fields.

We also identify that there is a vertical resolution threshold where the vertical velocity component is highly affected in deep water conditions. When the number of the layer is lower than a vertical resolution threshold, the wave phase error substantially increases over simulation time, and it becomes higher with a reduction of the number of layers. The vertical resolution threshold is less restrictive using the *FSFS* approach (10 layers) in comparison with the *CBA* approach (20 layers). In shallow water conditions, the vertical resolution seems to have a non-significant impact on wave frequency dispersion. The *FSFS* approach had satisfactory phase representation to 5 layers or more, and also had an adequate representation of nonlinear behavior, similar to previous work [20,23,24,35]. In summary, the *CBA* approach may obtain similar results to the *FSFS* approach if a feasible vertical resolution is used. For instance, the 20-layers *CBA* simulation has similar results to the 8-layers *FSFS* simulation, although with a substantial gain in computational cost.

When compared with the solutions of previous works [7–12], the proposed approach applied at the free-surface has (*i*) low cost of implementation and (*ii*) medium performance. The FSFS approach only required a simple local boundary treatment, and it does not change the vertical moment numeric discretization, as the existing methods do. Despite the numerical improvement and simple computational implementation, we observed a limitation imposed by the vertical resolution (number of layers) on the model accuracy, which makes a medium performance model. As the vertical discretization influences directly the accuracy of horizontal and vertical velocities (see Equations (17), (19), (22) and (25)), the proposed solution did not have good performance using a reduced number of layers (less than 10).

Previous works solution had similar satisfactory results with only 2 layers [7–9,12], and 4 layers [10,11] for the same benchmarks, with CPU time less than 0.4 s for these scenarios. Although, the proposed method had similar mean CPU time when compared with the same benchmark setup (10-layers simulation), 1.54 s (Intel Xenon 3.7 GHz), 1.83 s (Pentium 4 2.0 GHz) [8] and 2.02 s (i7 2.93 GHz) [11].

Moreover, the analysis of the results clarifies some questions related to the vertical resolution issue for this kind of cell-centered non-hydrostatic model. When using a treatment for non-hydrostatic pressure, the vertical resolution threshold can be more flexible, allowing to use approximately 2 times fewer layers to solve wave dispersion to acceptable accuracy, in a deep water situation. Although this paper indicated a threshold of 10 layers with the FSFS approach, more analysis is needed to establish a local relation between the number of layers and flow characteristics to solve the wave frequency dispersion properly.

## 5. Conclusions

The treatment of non-hydrostatic pressure boundary condition at the top layer is mandatory to the numerical model to reach satisfactory results compared to other models in the literature (e.g., [9,11,24,36]). The proposed FSFS approach is a low implementation cost method to improve the performance of non-hydrostatic models, which can reach satisfactory results with 2 times fewer layers than the CBA approach. Thus, reducing the mean computational cost of one time-step simulation by ca. 1.7 times, reaching similar results (CBA $\Delta t = 2.62$ s and FSFS $\Delta t = 1.22$ s).

Besides the improvements, the new boundary condition treatment is still limited by the vertical momentum discretization used, leading to a poor performance with low vertical resolution in a deep water situation (less than 10 vertical layers). In shallow water conditions, the vertical resolution seems to have a nonsignificant impact on wave frequency dispersion. Based on the difference between deep and shallow water conditions, more efforts are still needed to establish a local relation between the number of layers and flow characteristics to ensure that the model properly solves the wave frequency dispersion with minimum vertical resolution.

**Author Contributions:** Conceptualization, A.H.F.C., C.R.F.J., and D.M.-M.; methodology, A.H.F.C. and C.R.F.J.; software, A.H.F.C. and C.R.F.J.; validation, A.H.F.C. and C.R.F.J.; formal analysis, A.H.F.C. and C.R.F.J.; investigation, A.H.F.C. and C.R.F.J.; data curation, A.H.F.C. and C.L.B.C.; writing—original draft preparation, A.H.F.C.; writing—review and editing, A.H.F.C., C.R.F.J., C.L.B.C. and D.M.-M.; visualization, A.H.F.C., and C.L.B.C.; supervision, C.R.F.J. and D.M.-M.; project administration, A.H.F.C., C.R.F.J., and D.M.-M.; funding acquisition, D.M.-M. All authors have read and agreed to the published version of the manuscript.

**Funding:** This paper was funded by CNPq (Conselho Nacional de Desenvolvimento Científico e Tecnológico) through a research grant to Augusto Hugo Farias da Cunha (grant number: 149819/2017-0) and Cayo Lopes Bezerra Chalegre (grant number: 132643/2019-7).

**Acknowledgments:** We would like to thank the Global Lake Ecological Observatory Network (GLEON: www.gleon.org) for providing a venue and resources for lake science discussions.

**Conflicts of Interest:** The authors declare no conflict of interest. The funders had no role in the design of the study; in the collection, analyses, or interpretation of data; in the writing of the manuscript; or in the decision to publish the results.

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
