# Peer review of "Improvement of Non-Hydrostatic Hydrodynamic Solution Using a Novel Free-Surface Boundary Condition"

_water, doi:10.3390/w12051271_

Round 1
Reviewer 1 Report
This manuscript reported the authors' numerical work of simulating three-dimensional free surface flows where nonhydrostatic effect needs to be considered. The so-called frictional sublayer treatment was proposed and tested for different conditions, of which results were compared with other available method e.g. CBA. One of the main findings is that the proposed method has less computing time by achieving similar or even better outcomes which would be produced by the existing method.
This paper's work has been numerically based, the authors might consider to address the following areas to improve the quality of their presentation of the work:
- What is the significance of proposing their FSFS approach in addition to saving time? For example, is there any physical significance of making such approximation? Is there any physical implication for the sublayer? Why is a 20-layer CBA simulation only comparable to a 8-layer FSFS one and so on?
- Is there a clear concept about how the layers have been defined in simulations? Why different layers were used? Why 20 layers were regarded as a limit?
- The mathematical expressions in section 2.3 looks quite long in many cases, do you really need to present in this way?
- What do you mean by "experimental" in figure 6, table 5 etc.?
- You have provided a lot of detailed figures and data in tables, would it be possible to produce more concise or more persuasive results?
Author Response
Dear revisor,
Thank you very much for the contribution to the paper. Were very important to clarify some points and we hope to satisfactorly attend your suggestions.
- What is the significance of proposing their FSFS approach in addition to saving time? For example, is there any physical significance of making such approximation? Is there any physical implication for the sublayer? Why is a 20-layer CBA simulation only comparable to a 8-layer FSFS one and so on?
We add a clearer explanation at lines 172-176, in the proposed method presentation; and lines 317-321 at discussion sections. In summary: “That occurs because, in this approach, the non-hydrostatic pressure is estimated at the center of the layer, which can be further away from the surface as smaller the number of layers. When using the proposed FSFS condition, the special treatment that occurs in the upper layer minimizes these effects and provides a more physically consistent numerical solution, since this component disappears into real flows.”
That means the CBA simulation needs more vertical layers to represent the wave phase and amplitude properly, in this benchmark, 20 layers is comparable to an 8 layers FSFS result. This is expected, because a in 8 layers CBA approach, the non-hydrostatic pressure is defined far way from free-surface (at the center of the top layer) and the FSFS approach overcome this effect.
- Is there a clear concept about how the layers have been defined in simulations? Why different layers were used? Why 20 layers were regarded as a limit?
We add a clearer explanation at lines 194-196 at the Numerical Experiments definition. In Summary is:
“We used six different vertical resolutions (20 ~ 5 layers), as the most of the previous studies, and since previous analyses showed that more than 20 layers do not have substantial improvement over 20 layers non-hydrostatic solution”
- The mathematical expressions in section 2.3 looks quite long in many cases, do you really need to present in this way?
Yes, in section 2.3, the numerical discretization of the equations used in our model is presented. This implies referencing the variables by elements “i”, faces “j” and layers “k” depending on the variable used. In this way, the reader will be able to identify if the numerical approximation he uses is similar to ours and implement the proposed method in a didactic way. The discretization used in the paper is based initially on the TRIM model (Casulli, 2004) and is the update used on the IPH-ECO model (Fragoso Jr, 2009)
- What do you mean by "experimental" in figure 6, table 5 etc.?
We add a clearer explanation in lines 286 “obtained experimentally by Beji and Battjes (1993)”. This experimental results is frequently used to validate numerical models, see explanation between lines 205 – 212.
- You have provided a lot of detailed figures and data in tables, would it be possible to produce more concise or more persuasive results?
Thank you for the suggestion. The paper is about a numeric method improvement, and we separated the results in verification and validation steps for the proposed method, both essential to assure that the numerical improvement is physically consistent, conservative, and in fact, has better results than the previous methods. Figure 4 and the comparison between tables 2 and 3 summarize the verification results. Figure 6, with the comparison between station metrics in table 5, summarize the validation results. The implication of the results was evaluated in the discussion section that we hope to clarify some issues of the reader. Although we do not want to be redundant about the results in the discussion or conclusion sections, we recognize that it is dense reading.
Please see the attachment.
Best regards
Hugo Cunha

Reviewer 2 Report
The manuscript "Improvement of nonhydrostatic hydrodynamic solution using a novel free-surface boundary condition" is an interesting and valuable paper. The manuscript follows an excellent structures with a high quality and novelty. I attach my minor item.

Author Response
Dear revisor,
Thank you very much for your suggestions.
In the revised paper, we attend all comments and suggestions in the pdf file attached to this review.
Best regards
Hugo Cunha
Round 2
Reviewer 1 Report
The reviewer's comments have been addressed properly.
The novelty of this paper's work is one of the criticisms raised by the reviewer since the work is pure numerically based. The paper can be accepted for publication, but the authors should continuously to deepen their work with more physical and application insights in their future study.